# Application of Augmented Reality Interventions for Children with Autism Spectrum Disorder (ASD): A Systematic Review

**A. B. M. S. U. Doulah** [1,*] , **Mirza Rasheduzzaman** [1] , **Faed Ahmed Arnob** [2] , **Farhana Sarker** [3] , **Nipa Roy** [4] , **Md. Anwar Ullah** [5] **and Khondaker A. Mamun** [2,6,*]

1 Department of Electrical and Electronic Engineering, University of Liberal Arts Bangladesh (ULAB), Dhaka 1209, Bangladesh; mirza.rasheduzzaman@ulab.edu.bd
2 Advanced Intelligent Multidisciplinary Systems Lab (AIMS Lab), Institute of Research Innovation, Incubation and Commercialisation (IRIIC), United International University, Dhaka 1212, Bangladesh; faed.arnob60@gmail.com
3 CMED Health, Dhaka 1206, Bangladesh; farhana.sarker@cmedhealth.com
4 Institute of Natural Sciences, United International University, Dhaka 1212, Bangladesh; nipa@ins.uiu.ac.bd
5 Statistics and Informatics Division, Dhaka 1207, Bangladesh; anwar2023@yahoo.com.sg
6 Department of Computer Science and Engineering, United International University, Dhaka 1212, Bangladesh
\* Correspondence: abul.sayeed@ulab.edu.bd (A.B.M.S.U.D.); mamun@cse.uiu.ac.bd (K.A.M.)

**Abstract:** Over the past 10 years, the use of augmented reality (AR) applications to assist individuals with special needs such as intellectual disabilities, autism spectrum disorder (ASD), and physical disabilities has become more widespread. The beneficial features of AR for individuals with autism have driven a large amount of research into using this technology in assisting against autism-related impairments. This study aims to evaluate the effectiveness of AR in rehabilitating and training individuals with ASD through a systematic review using the PRISMA methodology. A comprehensive search of relevant databases was conducted, and 25 articles were selected for further investigation after being filtered based on inclusion criteria. The studies focused on areas such as social interaction, emotion recognition, cooperation, learning, cognitive skills, and living skills. The results showed that AR intervention was most effective in improving individuals' social skills, followed by learning, behavioral, and living skills. This systematic review provides guidance for future research by highlighting the limitations in current research designs, control groups, sample sizes, and assessment and feedback methods. The findings indicate that augmented reality could be a useful and practical tool for supporting individuals with ASD in daily life activities and promoting their social interactions.

**Keywords:** autism spectrum disorder; augmented reality; systematic review

## 1. Introduction

Autism spectrum disorder (ASD) is a neuro-developmental disorder determined by difficulties with social communication, restricted interests, and compulsive behavior [1]. Early detection of ASD is possible between the ages of 18 and 24 months; at this time, the diagnostic symptoms can be distinguished from regular developmental delays as well as other developmental disorders [2]. Globally, it is estimated that one in 100 children suffers from ASD. Significant advancements in international policy have significantly complemented the developments in autism research. Along with the policy changes brought about by the significant rise in global consciousness and campaigns, autism has been impacted by advancements in related fields including human rights, maternal and child health, and mental well-being [3–6]. The foundation and impetus for this development came from the United Nations Convention on the Rights of Persons with Disabilities (UNCRPD), which outlines key principles such as respect for dignity, freedom of choice, non-discrimination, full participation and inclusion in society, and accepting people with disabilities as a part of human diversity.

Each child with ASD is unique; it is possible that a technological solution that is beneficial for one child may not be effective for another. Therefore, researchers have begun to integrate a variety of technology for the betterment of children with autism with an aim of determining the best suited technologies for each individual. Technologies such as computer-based tools, virtual/augmented reality, mobile- and tablet-based applications, and robotics are currently viewed as propitious approaches for designing interventions for ASD, addressing a variety of goals including social and learning skills, on-task behavior, and challenging behaviors [7–19].

Early intensive behavioral intervention can ameliorate some of the symptoms of autism [7]. Nevertheless, the tools of certain classic media intervention approaches, such as video modeling (VM), are typically overly lengthy and lack an interactive mechanism [8–10]. It is difficult for ASD children to dynamically alter their cognitive focus and position. In recent years, information and communication technologies (ICTs) have been deployed extensively in the healthcare industry [11]. In the field of autism rehabilitation, technology-based interventions (TBI) include social robots [12], computer-based interventions [13,14], VR [15,16], tablet computers [17], and serious games [18,19].

As a new form of human–computer interaction (HCI) technology, AR has advanced significantly throughout the last few years to offer rich visual information and diverse interactive experiences by combining real scene information and virtual information [20–22]. AR presents fresh ideas for enhancing the learning experience [23]. There is evidence that autistic individuals are eager to analyze visual information and use electronic gadgets [24]. In addition, according to their parents, technology, such as tablets and smartphones, are significantly beneficial in treating these behavior issues [24,25]. Applications of mobile augmented reality (MAR) make the treatment more engaging [26,27] and improve the academic progress of children with autism [28].

In recent years, there has been a great deal of interest in examining the efficacy of technology-based strategies in training and teaching different skills, such as communication and social skills, academic skills, and information processing, to increase independence in children and adolescents with ASD. Marto et al. conducted a systematic literature review (SLR) of 16 primary papers on the use of augmented reality (AR) for the rehabilitation of people with ASD [29]. Only eight studies on MAR were considered. However, the authors of [30] examined ten preliminary studies published between 2012 and 2016 and made recommendations for future research and evaluation. Similarly, the collection contains only six studies on MAR. Adnan et al. [31] articulated AR's development and research prospects in the treatment of autism; however, the literature study was inadequate. Khowaja et al. [32] presented a systematic literature review of major works published in 2005–2018 on AR intervention in improving a variety of skills in children and adolescents with ASD and presented the research classification of ASD. However, the article did not include MAR and the research timeline must be revised. Berenguer et al. [33] assessed the efficacy of AR technology on ASD.

## 1.1. Research Question and Objects

Through a systematic review, this study examines how AR methods are used with children who have ASD. This study seeks to answer the following question by conducting a systematic review using the PRISMA methodology:

How effective are augmented reality interventions for the training and rehabilitation of individuals with autism spectrum disorder (ASD) in terms of social interaction, emotion recognition, cooperation, learning, cognitive skills, and living skills?

Additionally, by systematically evaluating the existing literature, this study seeks to address the following research objectives:

- Systematically review the existing literature on augmented reality interventions for children with autism spectrum disorder (ASD);

- Evaluate the effectiveness of augmented reality interventions in rehabilitating and training individuals with ASD across various domains, including social interaction, emotion recognition, cooperation, learning, cognitive skills, and living skills;
- Identify the strengths and limitations of the current research designs, control groups, sample sizes, and assessment and feedback methods used in the reviewed studies.

### 1.2. Contributions and Organization

This article presents a systematic literature review on the application of AR interventions for children with autism spectrum disorder (ASD) with an aim to investigate the recent trends, potential, and future research in AR technologies for the purpose of autism spectrum disorder intervention. This SLR includes works in the time span from 2010 to 2022. A summary of this work's essential contributions are the following:

- We conducted a systematic search for studies evaluating the types of intervention on the ASD population and analyze the efficacy of AR intervention on a variety of functions, such as social and communication, emotion management, daily living, and cognitive skills;
- We discussed the limitations and future works of the existing research.

The remaining sections of the paper are structured as follows. Section 2 discusses the linked concepts first. Then, we present the methodology in Section 3 before delving into the detailed results in Section 4. In Section 5, the discussion and limitations are presented. In Section 6, we conclude the paper with suggestions for future research.

## 2. A Brief Overview of The Related Concepts

### 2.1. Autism Spectrum Disorder

The fifth edition of the Diagnostic and Statistical Manual of Mental Disorders (DSM-5) defines autism spectrum disorder (ASD) as a condition characterized by deficits in two core domains: (1) social communication and social interaction and (2) restricted repetitive patterns of behavior, interests, and activities. Since 2013, the DSM-5 has included Asperger's disorder, childhood disintegrative disorder, Rett's disorder, and a number of other related diseases such as ASD. Despite this, many researchers continue to interchange Asperger's syndrome with ASD. According to a study conducted by the National Institute of Health (NIH) of the United States published in June 2018, 2.41 percent of American kids are diagnosed with an autism spectrum disorder. This represents a 0.94 percent increase from 2010. In Bangladesh, two community studies [34,35] conducted in 2005 and 2009 revealed that the statistics of children with autism were 0.2 and 0.84/1000 children, respectively.

### 2.2. Augmented Reality

Given the current technical advancements, AR technology is increasingly expanding into a range of fields, including gaming, travel, leisure, business, medicine, and education. Aggarwal and Singhal [36] define AR as the superimposition or augmentation of digital images onto real-world objects using a variety of AR technologies. The authors identified four distinct types of augmented reality: marker based, marker-less, projection based, and superimposition based. Marker-based augmented reality, also referred as image recognition, creates the output using a camera and a visual marker. Markers may consist of a quick response (QR) code, a two-dimensional (2D) code, a paper-based trigger image, or a physical object, provided that the camera can detect it. Marker-less augmented reality, also termed location-based augmented reality, utilizes a global positioning system (GPS) to provide location-based data. AR that relies on projection casts artificial light onto real-world objects. It is employed to project a three-dimensional (3D) interactive hologram. Another form is superimposition-based augmented reality, which replaces the original perspective of an object with an enhanced one, either partially or entirely. AR is contrasted with virtual reality (VR), another prominent immersion-based technology, in that digital information is augmented in the real world. In contrast, VR immerses the individual in an entirely virtual world.



Due to the proliferation of user-friendly and cost-effective mobile applications, augmented reality is also becoming more prevalent in education [37]. AR technology may be utilized to create interactive learning environments for children with ASD and intellectual development disability (IDD), allowing them to visualize complex topics and master a range of complicated activities in a visual environment [33,38]. The camera of a mobile device can be used to explore AR platforms while using AR. One can scan an image or marker, for instance, to display and manipulate digital information in a three-dimensional (3D) mode (e.g., a 3D visual of a cell will appear, which the user can touch or turn) or to display detailed information on an object (e.g., additional reading, video, or audio components will appear) [39]. AR technology has been determined to be an excellent instructional technology platform for assisting children with a wide range of impairments to acquire a variety of skills [40].

## 3. Methodology

### 3.1. Search Strategy and Data Sources

To perform this literature review, we sifted through seven scientific article databases: IEEE Xplore Digital Library, ACM Digital Library, Science Direct, Scopus, PubMed, Sage, and Web of Science. While analyzing these resources, we only considered documents relevant to computer-related categories, such as technology, engineering, and computer science, disallowing medical and chemical disciplines. In addition, we chose articles that were published between January 2010 and November 2022, spanning thirteen years.

### 3.2. Search Strings

We devised the search strings based on the topics pertinent to our systematic literature review. We selected a list of specific keywords, including "Autism Spectrum Disorder", "Augmented Reality", "Inclusive Digital Technologies", and "Smartphone" that would be relevant in answering our study questions. These strings focused on finding studies that studied or experimented with AR with ASD patients, taking into account user experience, accessibility, and region of experimentation. In Table 1, we list the search strings employed by the selected databases.

**Table 1.** Keywords.

| Category 1 | Category 2 | Category 3 |
|---|---|---|
| Autism | Augmented Reality | Mobile |
| Autism Spectrum Disorder | AR | Tablet |
| ASD | Mobile Augmented Reality | Smartphone |
| Autistic | Augmented | Toolkit |
| Autistic Children | Inclusive Digital Technologies | Smartglass |

The search was conducted in order to match the following logical expression: Title/Keywords/Abstract contains ("Autism" OR "Autism Spectrum Disorder" OR "ASD" OR "Autistic" OR "Autistic Children") AND ("Augmented Reality" OR "AR" OR "Mobile Augmented Reality" OR "Augmented" OR "Inclusive Digital Technologies") AND ("Mobile" OR "Tablet" OR "Smartphone" OR "Toolkit" OR "Smartglass").

### 3.3. Inclusion and Exclusion Criteria

We incorporated the requirements illustrated in Table 2 in order to address the research queries based on the selected publications and gain a comprehensive understanding of the designs we are dealing with.

**Table 2.** Inclusion and Exclusion Criteria.

| Inclusion Criteria | Exclusion Criteria |
|---|---|
| In1: Studies published between January 2010 and November 2022, spanning the last ten years | Ex1: Studies with an exclusively medical focus or a focus on the diagnosis of autism spectrum disorder |
| In2: Journal articles | Ex2: Not peer-reviewed (i.e., commentaries, letters to the editor, opinion articles, lectures) |
| In3: Studies with a focus on AR technology for ASD patients | Ex3: Studies that involve AR but do not include ASD exclusively |
| In4: Studies related to the usage of technology | Ex4: Studies that take into account usability and accessibility in contexts where technology is not involved |
| In5: Authors reported intervention impact, generalization, or maintenance | Ex5: Descriptive case studies with no reported research design or participant outcome. |
| In6: Empirically based studies employing single-subject, qualitative, quantitative, or mixed methodologies | Ex6: Studies that do not directly attempt to aid individuals with autism spectrum disorder, but rather their caregivers |

*3.4. Data Extraction*

Using a systematic approach for data extraction, the whole text of each selected article was thoroughly evaluated in order to extract key information. The data extraction included the following domains: study design, study characteristics, method/algorithm employed, significant findings, region of experimentation, and assessment methodology. Table 3 illustrates the domains of each paper.

**Table 3.** Characteristics of Included Studies.

| No. | Study | Study Design | Study Purpose | Functional Skills/Observed Improvements | No. of Participants/Consent Taken | Groups (Sex/Age) | Geographic Location/ Ethnicity | Duration of AR |
|---|---|---|---|---|---|---|---|---|
| 01. | Hashim (2022) [41] | Design and development | Assist children with mild autism in acquiring English vocabulary | Learning/significant improvements in learning | Total: 6 (M:5, F:1)/yes | Mildly ASD children; age: 5–12 years | Malaysia/Malaysian | 15–20 h in total |
| 02. | Nekar (2022) [42] | Single group;pre-test–post-test | Evaluate multiplayer games with dual-task exercise employing AR and a personal health record (PHR) system for autistic children's social skills and cognitive function | Social skills/promising improvement in social skills | Total: 14 (M:11, F:3)/yes | ASD children; age: 6–16 | South Korea/Korean | 3 h in total |
| 03. | Perez (2022) [43] | Multiple baseline single-subject experimental design | Improve responding to joint attention (RJA) skills such as gaze tracking and pointing | Joint attention/moderate engagement | Total: 6 (M:5, F:1)/did not mention | ASD children; age: 3–8 years old | Spain/Spanish | 30 min per session |
| 04. | Root (2022) [44] | Program evaluation | Analyze the effectiveness of a multi-component intervention using AR and modified schema-based instruction (MSBI) | Living skills/moderate improvements | Total: 4/did not mention | Age: 21 | USA | Did not mention |

**Table 3.** *Cont.*

| No. | Study | Study Design | Study Purpose | Functional Skills/Observed Improvements | No. of Participants/Consent Taken | Groups (Sex/Age) | Geographic Location/ Ethnicity | Duration of AR |
|---|---|---|---|---|---|---|---|---|
| 05. | Wang (2022) [45] | AR intervention study | Using support modules, dynamic video, and AR in (key partial video) KPV, enhance the communication of youngsters with ASD. | Social interaction, verbal/non-verbal communication, facial emotion recognition, attention | Total: 5 (M:4, F:1)/yes | 2 TD male 3 ASD (2 male and 1 female); mean age: 7 | China/Taiwanese | Did not mention |
| 06. | Lee (2021) [28] | Multiple baseline across single-subjects design | Help children with autism recognize social greetings and body gestures so they can respond appropriately | Social reciprocal skills/improvement observed | Total: 3 (M:2, F:1)/yes | ASD children; age: 7–9 years old (mean age = 8.1 years) | Taiwan/Taiwanese | Did not mention |
| 07. | Luca (2021) [46] | Single-subject design | Determine the feasibility and effectiveness of VR gadget in improving cognitive and behavioral skills of an ASD-affected boy | Cognitive and behavioral skills/indeterminate | Total: 1 (M:1)/did not mention | Male; age: 16 | Italy/Italian | 40 min per session |
| 08. | Zheng (2021) [47] | Feasibility study | Interactive AR coaching system, CheerBrush, to improve the tooth-brushing skills of children with ASD | Living skills/improved living skills | Total: 12/did not mention | (6 with ASD, 6 TD); 3–6 years | USA/American | Did not mention |
| 09. | Anto (2020) [48] | Pre-test–post-test | Employment of alphabet letters and numbers in AR environment and its influence on reaction time | Learning/significant improvement | Total: 96/yes | (48 with ASD, 48 TD); mean age | Brazil/Brazilian | Did not mention |
| 10. | Arpaia (2020) [49] | Case study | Design a system for ASD rehabilitation combining AR smart glasses and SSVEP brain–computer interface | Social interaction/significant improvement | Total: 3/Did not mention | Untrained ASD/ADHD children; Age: 8–10 years old | Italy/Italian | Did not mention |
| 11. | Lopez (2020) [50] | Program evaluation | To improve interaction, communication, and emotional intelligence in elementary school students, build a multiplayer game using marker-less MAR. | Social, emotional, social interaction/satisfactory results | Total: 38/yes | Age: 9–11 years (mean = 10.42, SD = 0.59) | Spain/Spanish | Did not mention |
| 12. | Kung (2019) [51] | Prototype, pre–post quasi-experimental design | A pre–post experimental research design named BLS (Basic Living Skills) E-courseware to help in teaching | Education/significant improvements | Total: 4/did not mention | Not mentioned | Malaysia/Malaysian | Did not mention |
| 13. | Sahin (2018) [52] | Single-case experimental design | Feasibility study of Empowered Brain Face2Face module, a social communication intervention using smart glasses | Social skills/moderate improvements | Total: 1 (M:1)/Yes | Fully-verbal ASD boy; age: 13 years | USA/American | 20 min daily for 3 weeks |

<div align="center">**Table 3.** *Cont.*</div>

| No. | Study | Study Design | Study Purpose | Functional Skills/Observed Improvements | No. of Participants/Consent Taken | Groups (Sex/Age) | Geographic Location/ Ethnicity | Duration of AR |
|---|---|---|---|---|---|---|---|---|
| 14. | Magrini (2019) [53] | Feasibility study | Create interactive games that are adjusted for a child's sensory preferences and level of challenge. Children's gross and fine motor capabilities, imitation skills, social interaction, and personal autonomy will all be improved by the prototype. | Social interaction, emotional intelligence/improved | Total: 10/yes | Subjects with ASD and dyspraxia; age: 6–10 years | Italy/Italian | 45 min per session |
| 15. | Singh (2019) [54] | Within-subject user study | Using AR applications to enhance learning experiences for less privileged autistic children. | Acquisition of certain skills/indeterminate | Total: 12 (M:6, F:6)/Did not mention | (6 boys, 6 girls); age: 9–12 years | India/Indian | Did not mention |
| 16. | Tang (2019) [55] | Pilot study | Design a lightweight AR-based mobile vocabulary study application for autistic children, particularly for outdoor and home use | Education/significant improvements | Study one: did not mention; study two: five/did not mention | Age: one under 5; another 6–8 years | China/Chinese | Did not mention |
| 17. | Vahabzadeh (2018) [56] | Short-term, uncontrolled pilot study | Empowered Brain, an AR smart glasses tool for behavioral and social communication, was used to assess ADHD-related symptom changes in ASD children and adolescents. | Social skills/significant improvements | Total: 8 (M:7, F:1)/yes | ASD individuals (male to female ratio of 7:1); mean age: 15 years | USA/American | Did not mention |
| 18. | Kurniawan (2018) [57] | Program evaluation | Examine the use of multimedia PECS (Picture Exchange Communication System)-based AR as an alternative learning tool for autism children's communication training | Social interaction/significant improvements | Total: 12/did not mention | Autistic children | Indonesia/Indonesian | Did not mention |
| 19. | Syahputra (2018) [58] | Pilot study | Design an AR system using leap motion controller to help train the focus of autistic individuals | Emotional intelligence/indeterminate | N/A/did not mention | N/A | Indonesia/Indonesian | N/A |
| 20. | Chen (2016) [59] | Single-subject with multiple baseline design | Improve ASD children' attention to nonverbal social cues | Social-emotional reciprocity/improved | Total: 6 (M:5, F:1)/yes | 5 boys and 1 girl and Age: 11–13 years; mean: 11.53 years | Taiwan/Taiwanese | Did not mention |
| 21. | Cihak (2016) [60] | Group-experimental design | Evaluate AR intervention in teaching a chain task to elementary students with ASD | Living skills/indeterminate | Total: 3 (M:3)/Yes | Male with ASD | USA/American | Did not mention |

**Table 3.** *Cont.*

| No. | Study | Study Design | Study Purpose | Functional Skills/Observed Improvements | No. of Participants/Consent Taken | Groups (Sex/Age) | Geographic Location/ Ethnicity | Duration of AR |
|---|---|---|---|---|---|---|---|---|
| 22. | Hosseini (2016) [61] | Quasi-experimental study | PECS method implemented, and program was for training and orientation of objects of the real world | Communication skills/indeterminate | Did not mention/N/A | 6–11 years | Iran/Iranian | Did not mention |
| 23. | McMahon (2016) [62] | Multiple probes across behaviors design | Assess AR intervention for teaching science vocabulary to college students with intellectual disabilities and ASD. | Education/improved navigation | Total: 4 (M:1, F:3)/yes | Three students with ID and one with ASD; age: 19–25 years | USA/American | Did not mention |
| 24. | Bai (2014) [22] | Within-subject experiment | Design an interactive method to visualize pretense in an open-ended play environment using AR | Emotional intelligence using games/significant improvement | Total: 12 (M:10, F:2)/N/A | ASC; 10 male and 2 female; 4–7 | UK/British | Did not mention |
| 25. | Escobedo (2014) [63] | Deployment study | Help teachers lessen the burden and workload of administering therapies by allowing children to use tangible objects | Education/enables multitasking abilities | Total: 21/did not mention | (7 teachers, 14 students); age of students (3–7, m = 5.08, sd = 0.90) | Mexico/Mexican | Did not mention |

## 4. Results

### 4.1. Data Selection

A summary of the selected articles is presented in Figure 1. It illustrates a PRISMA diagram and eventually selected articles for the review after the primary selection. We identified 4591 articles, out of which 501 were from IEEE Xplore, 1101 were from PubMed, 1120 were from Science Direct, 401 were from Web Science, 205 were from Scopus, 256 were from ACM library, 152 were from SpringerLink, 348 were from Sage, and 507 were from Google Scholar.

A total of 1084 duplicate papers were eliminated from the searches. The identified articles underwent a manual screening process based on their title and abstract. Subsequently, the articles were further screened based on their relevance in addressing the research questions. Following this rigorous screening process, out of the initial 3507 articles, only 25 articles met the inclusion criteria and were deemed suitable for further analysis in this systematic review. The study examined and collated the essential data from eligible studies, including the identity of the lead author, year of publication, characteristics of the participants, study design, and region of the experiment.

### 4.2. Descriptive Characteristics of The Selected Studies

#### 4.2.1. System Protocol Design

From the 25 studies reviewed, two studies [52,56] implemented empowered brain integration Google Glass and Face2Face module and four studies [41,54,57,63] developed their own android applications. The study [44] used a calculator app, HP Reveal, and AR markers using HP reveal. A recent study [51] implemented augmented reality animation, and some other studies [42,44,45,53,60] used different AR development applications such as HP Reveal, Unity Vuforia, Kinect, AR Kit, MAKAR, and Aurasma. In [48], the authors introduced a computer game named "MoviLetrando". The authors in [55] proposed and implemented deep learning models and simple, lightweight applications. Most of the studies reviewed [44,51,57,60,64] used the camera of tablets and smartphones. Two of the reviewed studies [52,56] used Google Glasses, whereas two other studies [45,54] used

a computer monitor as the AR setup. Some other studies [41,44,45,54] used flashcards, worksheets, picture cards, and Tangram puzzles in the learning process.

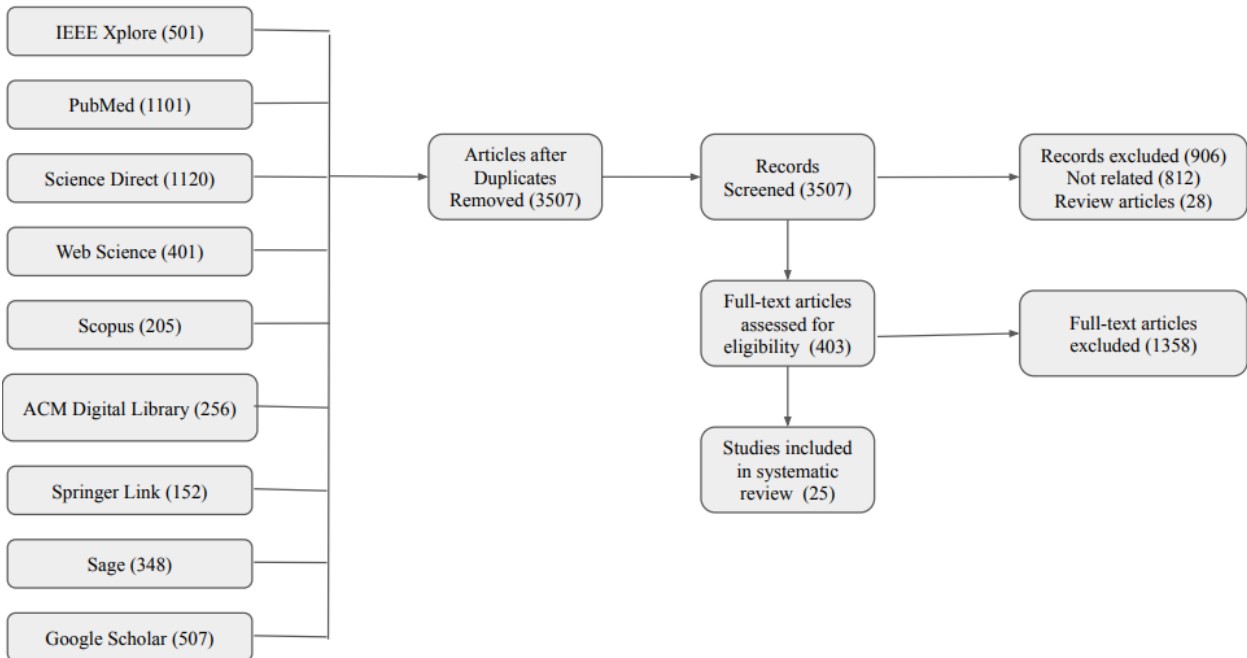

**Figure 1.** Flow chart of the study selection procedure.

4.2.2. Target Skills

A plurality of the studies (8 out of the 25) focused on improving social skills, particularly social interaction and reciprocity. Meanwhile, three studies aimed to enhance daily living skills such as teeth brushing, and two others centered on improving focus and attention; illustrated in Figure 2. Three additional studies specifically addressed behavior modification. Finally, a total of seven studies aimed to improve literacy and language skills, specifically by increasing vocabulary.

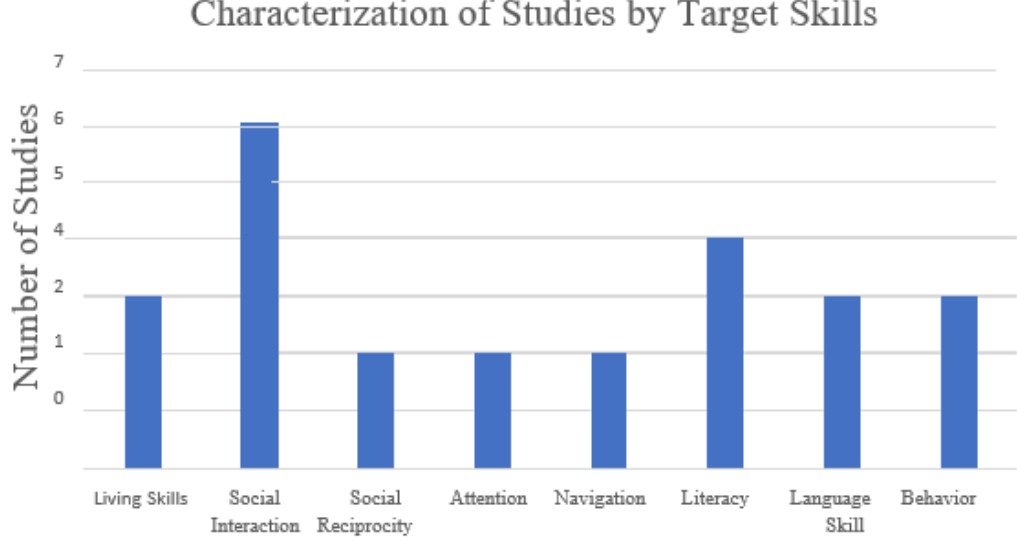

**Figure 2.** Characterization of studies by target skills.

### 4.2.3. Participant Characteristics

From the observation of Figure 3, Most of the studies included in this analysis involved participants whose ages ranged from 6 to 20 years old. Among all studies, 10 studies had a mean age between 6 and 9 years, and seven studies had a mean age between 10 and 20 years. Only three studies had a mean age below 5 years, and two studies had a mean age above 20 years old. Geographical location of the studies were also taken in account to illustrate the regions where the studies took place, it can been in Figure 4. Table 4 represents the assessment in which the characteristics of the studies were performed. This focuses on the inclusion criteria and the possible covariates of the selected studies.

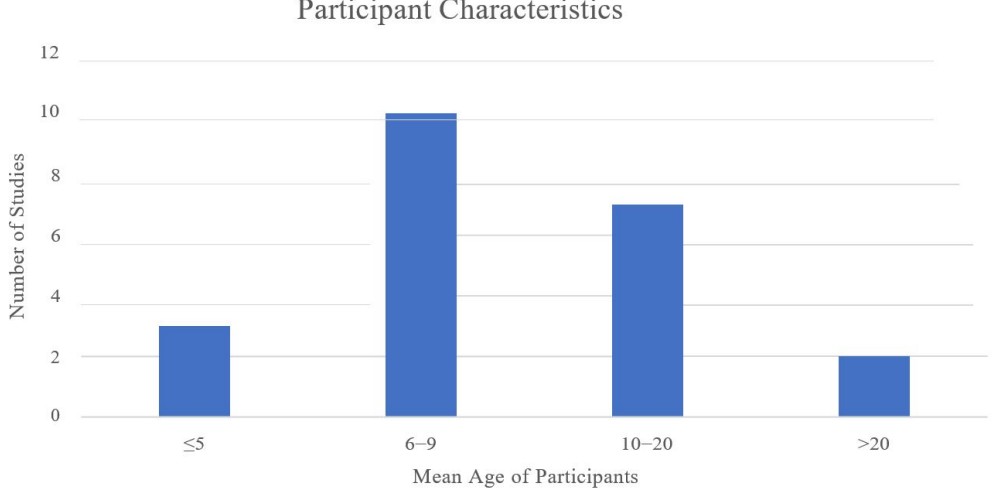

**Figure 3.** Participants Characteristics.

### 4.2.4. Region of Experimentation

A plurality of the studies included were performed in the United States (n = 6). Other studies were conducted in the UK, Iran, Malaysia, Indonesia, Mexico, and India.

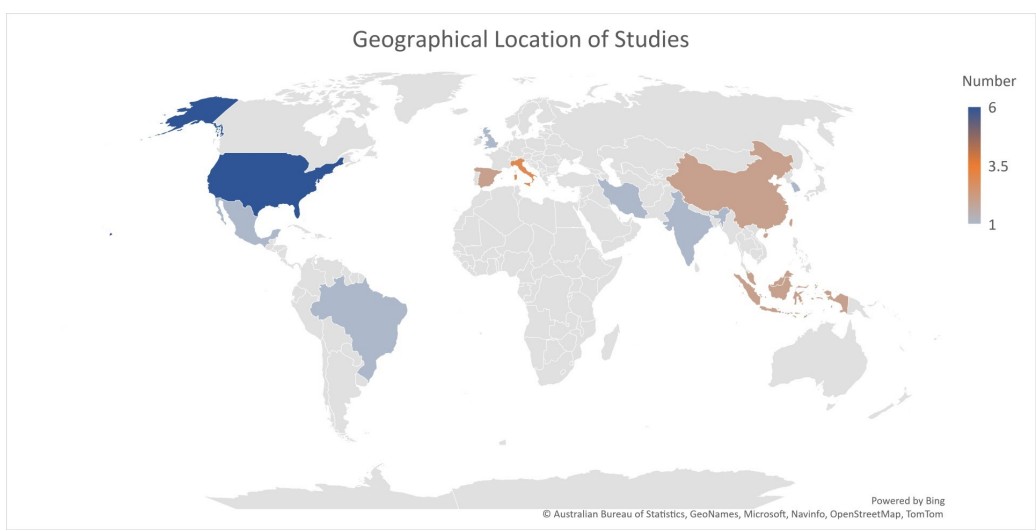

**Figure 4.** Region of Experimentation.

**Table 4.** Assessment Characteristics of Included Studies.

| No. | Study | Assessment of Inclusion Criteria | Possible Covariates |
|---|---|---|---|
| 01. | Hashim (2022) [41] | N/A | N/A |
| 02. | Nekar (2022) [42] | (1) Having been diagnosed with autism, (2) the ability to see, hear, and understand basic instructions, (3) the ability to read and understand Korean (the main language used in the game contents) | N/A |
| 03. | Perez (2022) [43] | N/A | N/A |
| 04. | Root (2022) [44] | (1) Educational or medical classification of ASD, (2) enrolled in a public school transition program, (3) researcher observation of prerequisite skills via screening tool. | N/A |
| 05. | Wang (2022) [45] | N/A | N/A |
| 06. | Lee (2021) [28] | (1) A clinical diagnosis of ASD based on DSM-IV-TR criteria, (2) no other specific disabilities, (3) not taking medications for physician or self-diagnosed illnesses, (4) no physician-diagnosed comorbidities, (5) not undergoing any other therapies at the time of the testing, (6) an FIQ > 90. | N/A |
| 07. | Luca (2021) [46] | N/A | N/A |
| 08. | Zheng (2021) [47] | (1) Height of at least 3 feet for successful Kinect recognition; (2) age 3–6 years; and, for children with ASD, (3) a documented diagnosis of ASD by a licensed clinical provider | N/A |
| 09. | Anto (2020) [48] | N/A | N/A |
| 10. | Arpaia (2020) [49] | N/A | N/A |
| 11. | Lopez (2020) [50] | N/A | H0h-P2Obs1-Sustaining mutual understanding, H0i-P2Obs2-Dialogue management, H0j-P2Obs3-Information pooling, H0k-P2Obs4-Reaching consensus, H0l-P2Obs5-Task division, H0m-P2Obs6-Time management, H0n-P2Obs7-Technical coordination, H0o-P2Obs8-Reciprocal interaction, H0p-P2Obs9-Individual task orientation. |
| 12. | Kung (2019) [51] | N/A | Basic Living Skills (BLS) e-courseware prototypes (Augmented Reality Animation) and Basic Living Skills (Static graphic) |
| 13. | Sahin (2018) [52] | N/A | N/A |
| 14. | Magrini (2019) [53] | N/A | N/A |
| 15. | Singh (2019) [54] | N/A | Training Time, solve time, correct placements, subjective questionnaire. |
| 16. | Tang (2019) [55] | N/A | N/A |
| 17. | Vahabzadeh (2018) [56] | N/A | N/A |
| 18. | Kurniawan (2018) [57] | N/A | N/A |
| 19. | Syahputra (2018) [58] | N/A | N/A |
| 20. | Chen (2016) [59] | (1) A clinical diagnosis of ASD based on DSM-IV-TR criteria, (2) not undergoing any other therapies/medication at the time of the testing, (3) no other disabilities, (4) an FIQ > 90. | N/A |
| 21. | Cihak (2016) [60] | N/A | N/A |
| 22. | Hosseini (2016) [61] | N/A | N/A |
| 23. | McMahon (2016) [62] | N/A | N/A |
| 24. | Bai (2014) [22] | N/A | N/A |
| 25. | Escobedo (2014) [63] | N/A | N/A |

### 4.2.5. Quality Assessment of the Articles

The quality assessment of the included 25 papers was performed using Critical Assessment of Structure Prediction (CASP) [65]. The CASP checklist has 10 questions divided in three sections: Section A, B, and C. Tables 5 and 6 illustrates the CASP. Furthermore, Table 7 cumulatively illustrates the key features of the included studies. It shows the software and hardware that are used also the assessment method followed by the authors. The following questions of the sections have assisted in assessing the quality of the papers

**Table 5.** Quality Assessment Using CASP-Section A.

| No. | Study | Did the Paper Address a Clearly Focused Question? | Did the Authors Look for the Right Type of Papers? | Do You Think All the Important, Relevant Studies Were Included? | Did the Paper's Authors Do Enough to Assess Quality of the Included Studies? | If the Results of the Paper Have Been Combined, Was It Reasonable to Do So? |
|---|---|---|---|---|---|---|
| 01. | Hashim (2022) [41] | Yes | Yes | Yes | Yes | Yes |
| 02. | Nekar (2022) [42] | Yes | Yes | No | Indeterminate | Yes |
| 03. | Perez (2022) [43] | Yes | Yes | Yes | Yes | Yes |
| 04. | Root (2022) [44] | Yes | Yes | Yes | Yes | Yes |
| 05. | Wang (2022) [45] | Yes | Yes | Yes | Yes | Yes |
| 06. | Lee (2021) [28] | Yes | Yes | Yes | Yes | Yes |
| 07. | Luca (2021) [46] | Yes | Yes | No | Yes | Yes |
| 08. | Zheng (2021) [47] | Yes | Yes | Yes | Yes | Yes |
| 09. | Anto (2020) [48] | Yes | Yes | Yes | Yes | Yes |
| 10. | Arpaia(2020) [49] | Yes | Yes | Yes | Yes | Yes |
| 11. | Lopez (2020) [50] | Yes | Yes | Yes | Yes | Yes |
| 12. | Kung (2019) [51] | Yes | Yes | Yes | Yes | Yes |
| 13. | Sahin (2018) [52] | Yes | Yes | Yes | Yes | Yes |
| 14. | Magrini (2019) [53] | Yes | Yes | Yes | Yes | Yes |
| 15. | Singh (2019) [54] | Yes | Yes | Yes | Yes | Yes |
| 16. | Tang(2019) [55] | Yes | Yes | No | Indeterminate | Indeterminate |
| 17. | Vahabzadeh (2018) [56] | Yes | Yes | Yes | Yes | Yes |
| 18. | Kurniawan (2018) [57] | Yes | No | No | Yes | Indeterminate |
| 19. | Syahputra (2018) [58] | No | No | No | Indeterminate | Indeterminate |
| 20. | Chen (2016) [59] | No | Yes | Yes | Yes | Yes |
| 21. | Cihak (2016) [60] | Yes | Yes | Yes | Yes | Yes |
| 22. | Hosseini (2016) [61] | Yes | Yes | Yes | Yes | Yes |
| 23. | McMahon (2016) [62] | Yes | Yes | Yes | Yes | Yes |
| 24. | Bai (2014) [22] | Yes | Yes | Yes | Yes | Yes |
| 25. | Escobedo (2014) [63] | Yes | Yes | No | No | No |

**Table 6.** Quality Assessment Using CASP-Section A and B.

| No. | Study | What Are the Overall Results of the Paper? | How Precise Are the Results? | Can the Results Be Applied to the Local Population? | Were All Important Outcomes Considered? | Are the Benefits Worth the Harms and Costs? |
|---|---|---|---|---|---|---|
| 01. | Hashim (2022) [41] | Did not mention clearly | Did not mention | Indeterminate | Indeterminate | Indeterminate |
| 02. | Nekar (2022) [42] | Despite no statistically significant result in social communication and restricted interests and repetitive behavior, a decrease in the mean was observed when compared to the baseline data. | $p < 0.05$ | Yes | Yes | Yes |

**Table 6.** *Cont.*

| No. | Study | What Are the Overall Results of the Paper? | How Precise Are the Results? | Can the Results Be Applied to the Local Population? | Were All Important Outcomes Considered? | Are the Benefits Worth the Harms and Costs? |
|---|---|---|---|---|---|---|
| 03. | Perez (2022) [43] | An overall PAND of 98% was obtained for v2 and an overall PAND of 96% was measured for v3. This shows that the intervention was highly effective (PAND > 90%) for enhancing the abilities of gaze following and pointing to the target object in six autistic children. | $p < 0.01$ | Indeterminate | Yes | Yes |
| 04. | Root (2022) [44] | The combination of MSBI and video-based instruction delivered via AR helped four adult students with ASD improve in their ability to solve percentage of change problems (i.e., tip) | Did Not Mention | Yes | Yes | Yes |
| 05. | Wang (2022) [45] | ASD improved from baseline 1 at 83% when using the AOM to 98% at intervention 2 when using KPV with AR | $p < 0.05$ | Indeterminate | Yes | Indeterminate |
| 06. | Lee (2021) [28] | Baseline mean: 19.16% | $p < 0.05$ | Yes | Yes | Yes |
| 07. | Luca (2021) [46] | Significant increase in attention processes | Did not mention | Indeterminate | Indeterminate | Yes |
| 08. | Zheng (2021) [47] | ASD Baseline: 98.89 | The HR and SCL differences of ASD group reached and was near statistically significant, respectively | Indeterminate | Yes | Yes |
| 09. | Anto (2020) [48] | ASD Total Points: 54.5 | $p < 0.05$ | Indeterminate | Yes | Yes |
| 10. | Arpaia (2020) [49] | Positive feedback on the device acceptance and attentional performance were offered after tests on three ASD patients (with three different CGI scores) between 8 and 10 years | Did not mention | Yes | Yes | Yes |

**Table 6.** *Cont.*

| No. | Study | What Are the Overall Results of the Paper? | How Precise Are the Results? | Can the Results Be Applied to the Local Population? | Were All Important Outcomes Considered? | Are the Benefits Worth the Harms and Costs? |
|---|---|---|---|---|---|---|
| 11. | Lopez (2020) [50] | A total of 95% of the children participated actively in the search for the solution to end the game | $p < 0.05$ | Yes | Yes | Yes |
| 12. | Kung (2019) [51] | Improvement in academic performance | Did not mention | Yes | Indeterminate | Yes |
| 13. | Sahin (2018) [52] | SRS measure baseline raw score: 92 | Did not mention | Yes | Yes | Yes |
| 14. | Magrini (2019) [53] | Children embraced experiments, showing improved attention, and response times | Did not mention | Yes | Indeterminate | Yes |
| 15. | Singh (2019) [54] | More training time compared to desktop and in-person | Did not mention | Yes | Indeterminate | Yes |
| 16. | Tang (2019) [55] | Did not mention clearly | N/A | Indeterminate | Indeterminate | Indeterminate |
| 17. | Vahabzadeh (2018) [56] | Baseline mean: 5.5 | Did not mention | Yes | Yes | Yes |
| 18. | Kurniawan (2018) [57] | The increase achieved an average of 76% in the communication skills of children before treatment. | Did not mention | Yes | Yes | Yes |
| 19. | Syahputra (2018) [58] | With an average percentage of 83% on a Likert scale, respondents strongly agree that the app can train the child's focus. | Did not mention | Indeterminate | Indeterminate | Indeterminate |
| 20. | Chen (2016) [59] | baseline range: 86.66–94.28% | Yes | No | Yes | Yes |
| 21. | Cihak (2016) [60] | Students' independent performance immediate increased when augmented reality was introduced with 98% non-overlapping data. | Did not mention | Indeterminate | Indeterminate | Yes |
| 22. | Hosseini (2016) [61] | Data obtained results using Wilcoxon test of 0.007 | $p < 0.05$ | Yes | Yes | Yes |

**Table 6.** *Cont.*

| No. | Study | What Are the Overall Results of the Paper? | How Precise Are the Results? | Can the Results Be Applied to the Local Population? | Were All Important Outcomes Considered? | Are the Benefits Worth the Harms and Costs? |
|---|---|---|---|---|---|---|
| 23. | McMahon (2016) [62] | Baseline scores on the vocabulary assessments for the students indicated that the students had very low initial knowledge of the science vocabulary words across the three-word lists | Did not mention | Yes | No | Yes |
| 24. | Bai (2014) [22] | Did not mention clearly | Did not mention | Indeterminate | Indeterminate | Indeterminate |
| 25. | Escobedo (2014) [63] | Mobile Object Identification System helped teachers to attend to multiple students (n > 1) during the therapy | Did not mention | Yes | No | Yes |

**Table 7.** System Protocol Design.

| No. | Study | Software | Hardware | Setting | Phase | Duration | Method Followed/Feedback Method | Survey Questions Availability |
|---|---|---|---|---|---|---|---|---|
| 01. | Hashim (2022) [41] | Smartphone application, called 'AReal-Vocab' | Flashcards containing words | Home and Classroom | Five 30 to 40 min sessions | 5 weeks | Kohan Cappa Analysis/ interviews and field notes | Yes |
| 02. | Nekar (2022) [42] | Kinect | Tablet | Local social welfare center | Baseline, intervention, post-intervention Survey | Three weeks | 3 Wilcoxon signed-rank test/System Usability Scale (SUS)-based survey | Yes |
| 03. | Perez (2022) [43] | Pictogram Room | LCD Projector, Interactive Digital Whiteboard, Computer, Kinect, and Samsung SCC-301P Video camera | School classroom | Pre-baseline phase, pre-assessments, baseline phase, learning phase, intervention phase, post-intervention phase, post-assessments, follow-up assessments | 12 Weeks | Social Communication Questionnaire, Leiter International Performance Scale, Autism Diagnostic Observation Schedule Second Edition (ADOS-2), Early Social Communication Scales (ESCS)/N/A | Yes |
| 04. | Root (2022) [44] | Calculator application, HP Reveal, and First-Then Visual Schedule HD | Worksheet and iPod | Classroom, Campus coffee and snack shop | Baseline, intervention, generalization, and maintenance | N/A | N/A | No |

**Table 7.** *Cont.*

| No. | Study | Software | Hardware | Setting | Phase | Duration | Method Followed/Feedback Method | Survey Questions Availability |
|---|---|---|---|---|---|---|---|---|
| 05. | Wang (2022) [45] | HP Reveal, Unity Vuforia, AR Kit, MAKAR, KPV | Paper-based picture cards, vocabulary textbooks, tablet | Teacher-selected computer classrooms in elementary schools | Two phases: the first phase comprises the A1 baseline and B1 treatment periods, and the second phase includes the C1 reversal and B2 treatment periods | 4 Months | t-test/N/A | No |
| 06. | Lee (2021) [28] | KST System | Computer | Classroom | Baseline, Intervention, Maintenance | 6 weeks | Kolmogorov–Smirnov test, Likert scale/Verbal | No |
| 07. | Luca (2021) [46] | N/A | Medical Device named BTS-N | Clinic | Two different trainings: CBT according to standardized approach, CBT in a VR environment (Total 48 sessions) | One month | Raven's Matrices test, Modified Little Bell Test, developmental test of visual–motor integration, Gilliam Autism Rating Scale/ Questionnaire | No |
| 08. | Zheng (2021) [47] | Virtual Avatar | Mechatronic toothbrush and Microsoft Kinect V2 Camera and Sensor, E4 | N/A | Baseline, pre-test, coaching, post-test | N/A | *p*-values/ Questionnaire, Likert Scale | Yes |
| 09. | Anto (2020) [48] | Computer game MoviLetrando | Laptop computer | Classroom | Pre-test, post-test | N/A | Mann–Whitney U test, Wilcoxon signed-rank test/N/A | No |
| 10. | Arpaia (2020) [49] | A SSVEP-based single-channel BCI | Smart glasses | Social interaction | N/A | N/A | N/A | No |
| 11. | Lopez (2020) [50] | EmoFindAR application | Smartphone | Classroom | Gender: did not mention | N/A | Questionnaire/ N/A | Yes |
| 12. | Kung (2019) [51] | Smartphone application | Smartphone and flashcards | Classroom | Pre-test and post-test | N/A | N/A/ Questionnaire | No |
| 13. | Shanin (2018) [52] | N/A | Empowered Brain Face2Face module | Classroom | N/A Baseline and intervention | 2 weeks | N/A/Social Responsiveness Scale2 (SRS-2) school-age form | No |
| 14. | Magrini (2019) [53] | Kinect SDK | Computer, Sensor | N/A | Four phases: repeat the movements, guess the movements, connect the dots and guess the card | 6 weeks | Time and score for each exercise/verbal | No |
| 15. | Singh (2019) [54] | Desktop-based Application | Tangram Puzzle and Desktop Computer and Webcam | Classroom | Training and Testing | N/A | Time required to solve a Tangram puzzle/ Questionnaire | Yes |
| 16. | Tang (2019) [55] | Deep learning platform, TensorFlow and simple lightweight mobile application | N/A | Special education school and university campus | N/A | N/A | No/verbal | No |

**Table 7.** *Cont.*

| No. | Study | Software | Hardware | Setting | Phase | Duration | Method Followed/Feedback Method | Survey Questions Availability |
|---|---|---|---|---|---|---|---|---|
| 17. | Kurniawan (2018) [57] | Mobile application | Smartphone | Classroom | Baseline, Intervention after 24 and 48 h | N/A | Qualitative, visual analysis/interview | No |
| 18. | Vahabzadeh (2018) [56] | Empowered Brain System | Google Glass | N/A | Baseline, Intervention after 24 and 48 h | 48 h | Social Communication Questionnaire (SCQ), ABC-H/N/A | No |
| 19. | Syahputra (2018) [58] | Leap Motion Controller | Camera, 3D object marker, Computer | N/A | N/A | N/A | Detection/Questionnaire | No |
| 20. | Chen (2016) [59] | Application | Tablet and AR-based video modeling storybook | School | Baseline, intervention, maintenance | 4 weeks | No/ questionnaires and interviews after each phase | No |
| 21. | Hosseini (2016) [61] | Mobile application | Smartphone | School | N/A | N/A | Wilcoxon test for data collection/N/A | No |
| 22. | Bai (2014) [22] | Goblin XNA | Webcam, Bluetooth keyboard, and play materials consisting of AR items (three foam blocks and a box with connected markers) and non-AR physical props (three cotton balls, two paper tubes, three popsicle sticks, three pen tops, three strings and a piece of cloth) | Home | N/A | N/A | CARS2, BPVS3, play frequency, play time/ questionnaire | No |
| 23. | Cihak (2016) [60] | AR application | iPod, toothbrush, paper cup | Classroom | Baseline, pretraining, AR intervention, maintenance | 9 weeks | Likert scale, percentage of steps/no | No |
| 24. | McMahon (2016) [62] | Mobile app Aurasma | Book | Computer lab in a university campus | Baseline, training, intervention | N/A | Number of vocabulary items defined and labeled correctly/ Likert-type survey | Yes |
| 25. | Escobedo (2014) [63] | MOBIS, an AR application | Smartphone | Specialized clinic | Pre-deployment (2 weeks), deployment (5 weeks), and post-deployment (1 week) | 8 weeks | Weekly interview; LSA and ANOVA/group interview, survey | No |

## 5. Discussion

This systematic literature review explores the potential of establishing a standard protocol in the field of technology to benefit children with autism spectrum disorder (ASD). The review analyzed twenty-five articles, published between 2010 and 2022, retrieved from various reputable databases. Participants' ages ranged from 6 to 20 years old. However, three studies [51,57,58] did not mention the age of the participants. The findings emphasized the importance of having a standard protocol for data collection and

the development of augmented reality (AR) technology tailored for children with ASD. Currently, AR researchers are creating impressive prototypes [18,21,28,48,51], but their implementation is hindered by a lack of communication between caregivers and developers. A crucial step towards progress involves fostering effective two-way communication between the researchers and the caregivers. Furthermore, government policy makers should actively engage in this field to support and empower children with ASD and their parents. The widespread adoption of this technology will heavily rely on the intervention and support of the government. Another objective of this review was to evaluate and summarize research that dealt with the efficacy of using AR programs in treating children and adolescents with ASD. Social skills, emotion recognition, daily living skills (brushing, finance, etc.), verbal and non-verbal communication, and learning can be distinguished as distinct areas of intervention when considering the capabilities of augmented reality technologies.

Social skills, the most obvious deficiency in children with ASD, were given the most attention in the examined AR study. Using a blend of real-world and virtual elements to replicate social environments allows for the training to occur in a safe, regulated, and customizable environment. For ASD treatment, the characteristics of this kind of intervention are particularly intriguing. In these articles, improvements in social–emotional reciprocity and emotional intelligence [28,47,50,53,59] were significantly observed. It is important to highlight that among the 25 selected papers, 18 of them documented notable enhancements in social-emotional reciprocity and emotional intelligence. In Table 3, we have illustrated the study purpose of each study and its outcome. Additionally, these approaches assist in the development of important social abilities such as initiation of play, social response, and facial expression perception. The studies [26,28] found that AR systems can positively improve understanding of social greeting behavior and learning appropriate responses to greetings. A few other studies [22,42] evaluated the feasibility of game content to assist autistic children with their social and emotional development. Other studies have been based on areas related to training in activities of daily living skills, teaching language, education and improving attention to individuals with ASD. Studies [41,48,51,55,62,63], demonstrate that using augmented reality can offer adjustable learning environments including videos, 3D images, and animation where people with special needs can engage in more fulfilling interaction activities. Children and adolescents with ASD could learn basic human skills such as tooth brushing and financial skills through AR intervention [44,47].

The majority of the included research suggests that children with ASD benefit from exposure to AR-based interventions. However, the majority of research utilized small sample sizes. Two studies [46,52] only comprised a single subject. Fourteen of the twenty-five studies had individuals under the age of ten. One study cited as [58] missed the number of participants. There was rarely any kind of comparison to either a group of healthy volunteers or patients who were receiving conventional therapies. If the control group does not exhibit the similar features or the same questionnaires are provided at distinct points in the intervention procedure, comparing the changes produced by an AR intervention could be challenging. Considering this, some of the findings from the research included in this study might be regarded as preliminary and having limited practical utility. Due to the limited sample size and lack of a control group, extending the results to the entire population impacted by this disorder is challenging. Another factor that makes it difficult to generalize the results is the sample's gender ratio. It is established that ASD affects boys more frequently than girls (the most recent research indicates a 4:1 ratio [2]); however, certain studies [46,52,60] are conducted exclusively with affected boys, which may limit the validity of the conclusions. Lastly, it should be mentioned that some research was conducted on children labeled with high-functioning autism or Asperger syndrome. Consequently, only results from this sub-sample should be evaluated, as they cannot be generalized to the remaining children with ASD.

One of the primary benefits of augmented reality is that it enables the simulation of real-world circumstances so that training may be undertaken in a safe, therapist-controlled environment. This issue is particularly intriguing when treatment should concentrate on

the training and enhancement of social skills, social interaction, communication, emotional reaction, and executive functions. In addition, this intervention style can be further extended to acquire various subject performance measures. This enables therapists to monitor and analyze the patient's progress and to apply feedback or possible task repetitions. Consequently, an intervention based on technology may incorporate multi-user apps in playful environments or everyday scenarios, which could be regulated and customized based on the intervention's goals. Despite the numerous potential benefits of augmented reality, it is recommended that practitioners do cost–benefit analyses to see whether there is a benefit to facilitating the specific interventions or intervention components under consideration with augmented reality. Notably, the majority of the research included in this evaluation still relied on the therapist to prompt or correct errors. This prompts the question of whether the integration of AR would lead to more effective and efficient therapies. It is also likely that there exist low-tech alternatives that achieve the same effects as augmented reality. Therefore, it would be useful to identify the qualities of the individuals for which AR technologies are most beneficial (e.g., cognitive ability, physical ability, etc.).

In accordance with the WHO Ethics Review Committee (ERC), when proposing research with human participants, an Informed Consent Form (ICF) must be included with each proposal to establish that the study participant has chosen to participate in the research voluntarily. If the research involves more than one group of participants, such as healthcare users and healthcare professionals, a separate, individualized informed consent form must be supplied for each group. This guarantees that each participant group receives the necessary information to make an informed decision. Each new intervention requires a different informed consent form for the same reason. Twelve of the research explicitly state that written agreement was obtained from the participants. However, in the other thirteen investigations, there was no statement of a signed consent form. In research initiatives, evaluation methods are also a major concern. Both survey research and qualitative research rely on human participation and require well-trained observers to ensure accurate and reliable results. Without proper training, the error rate can increase, and the usefulness and reliability of the measurement can decrease. To avoid the introduction of human errors, researchers should implement a comprehensive training program for observers prior to the intervention. This training should cover all aspects of the research process, including data collection and analysis, to ensure that observers are equipped with the necessary knowledge and skills to carry out their tasks accurately. Additionally, researchers can minimize the risk of human errors by conducting numerous calculations of the data or by using computer-assisted data gathering techniques. These methods can help to streamline the data collection process and reduce the potential for human error. In addition, the majority of research did not employ a particular method for evaluating the efficacy of AR interventions. Several studies [22,26,28,41–43,45,48,56,61,64] used a variety of established methods, including the Mann–Whitney U Test, Wilcoxon signed-rank Test, Social Communication Questionnaire (SCQ), ABC-H, CARS2, BPVS3, LSA, and ANOVA. Some research gathered input from participants, parents, or teachers through survey questionnaires, whereas others employed verbal feedback. A few other studies did not mention any such feedback method. Only four studies disclosed their questionnaire form in the papers.

Based on the reviews of selected articles, it is evident that future research in the field of AR for individuals with ASD should pay closer attention to comparing the various AR technologies employed because AR encompasses a wide range of technologies, some of which may potentially have compatibility issues, particularly concerning user immersion. Therefore, it is advisable that the researchers should shed light on AR technologies on their relative effectiveness, usability, and potential limitations.

Most of the studies included in this review were conducted in developed countries (United Nations Development Programme). Out of 25 studies, a total of 15 studies were conducted in the developed countries. Augmented reality (AR) technology has the potential to bring many benefits to both developed and developing countries; however, there are some limitations to consider [66]. In many developing countries, access to technology,

including smartphones and tablets, is limited. This can make it difficult for people to access and use AR applications. Internet connectivity in many developing countries is also limited, which can make it difficult for people to access and use AR applications that require a stable internet connection. Apart from that, in some areas, electricity is not available or is not reliable, which can make it difficult to use devices that require electricity, such as smartphones and tablets. Another limitation is that many AR applications are developed primarily in English, which can make them difficult to use for people in developing countries who speak other languages. Additionally, developing countries often have limited financial resources, which can make it difficult for people to afford to purchase technology such as smartphones and tablets, or to pay for internet connectivity. Finally, many developing countries may not have the necessary infrastructure or regulations to ensure the privacy and security of data collected or stored by AR applications. It is important to note that although there are limitations, researchers and practitioners are working to overcome these challenges and make AR technology more accessible and effective in developing countries.

Although this review has summarized and evaluated various studies, it is crucial to acknowledge that the effectiveness of interventions can vary widely from person to person. This individual variation presents a significant challenge when attempting to extrapolate data from individual studies to the entire ASD population. Therefore, it is important to provide emphasize the need for personalized, client-centered approaches in the treatment and support of individuals with ASD. This approach may lead to a better understanding of ASD as a spectrum and promote interventions aimed at enhancing the well-being and development of individuals with ASD.

## 6. Conclusions and Future Work

ASD, from its identification to its treatment, is a complicated condition; it does not appear in the same way in all children; its severity can be low, moderate, or severe; and according to many authors, it is a disorder with no cure and a lifetime duration. Because ASD manifests itself in children in a variety of ways and degrees, making them receptive to various stimuli, it seeks alternatives to the same behavior or reaction to the same stimulus. As a result, other particular areas must be targeted to strengthen their relationship with their environment. This study shows how effective AR intervention is at minimizing ASD-related deficiencies. The substantial effectiveness observed for activities of daily living could justify the use of AR therapies in clinical practice. For future research, the selection of participants, the type and duration of the intervention, and the choice of a measurement tool should incorporate more controlled features, and more effort should be devoted to follow-up assessments conducted weeks or months after the termination of the intervention to ensure that the effects of training are maintained. In our search, only peer-reviewed articles were included to eliminate a number of possibly relevant gray literature articles and conference papers. However, as our goal is to provide researchers and practitioners with constructive advice, it is essential to examine the quality of the studies. Future studies should aim to use the relevant study designs and procedures to close the gaps in the quality criteria. Heterogeneity of symptoms, individualized demands, and contextual circumstances are some critical features that are likely to be defined in future studies on autism. Furthermore, discussed as crucial AR features are multi-sensory integration, customization and flexibility, social interaction facilitation, generalization and transferability, and immersive and interactive experiences.

**Author Contributions:** Conceptualization, A.B.M.S.U.D. and M.R.; methodology, A.B.M.S.U.D.; software, F.A.A.; validation, F.S., N.R., M.A.U. and K.A.M.; formal analysis, A.B.M.S.U.D. and F.A.A.; investigation, A.B.M.S.U.D. and M.R.; resources, A.B.M.S.U.D.; data curation, A.B.M.S.U.D. and F.A.A.; writing—original draft preparation, A.B.M.S.U.D.; writing—review and editing, F.A.A.; visualization, F.A.A.; supervision, A.B.M.S.U.D. and M.R.; project administration, N.R., M.A.U. and K.A.M.; funding acquisition, F.S., N.R., M.A.U. and K.A.M. All authors have read and agreed to the published version of the manuscript.

**Funding:** This work was supported by Institute for Advanced Research (IAR) of United International University (UIU) in collaboration with Office of Faculty Research, University of Liberal Arts Bangladesh (ULAB) under Grant UIU-IAR-01-2022-SE-34.

**Data Availability Statement:** Not applicable.

**Conflicts of Interest:** The authors declare no conflict of interest.

## Abbreviations

The following abbreviations are used in this manuscript:

| | |
|---|---|
| AR | Augmented Reality |
| ASD | Autism Spectrum Disorder |
| GPS | Global Positioning System |
| HCI | Human-Computer Interaction |
| ICTs | Information and Communication Technologies |
| IoT | Internet of Things |
| MAR | Mobile Augmented Reality |
| SLR | Systematic Literature Review |
| SUS | System Usability Scale |
| PHR | Personal Health Record |
| TBI | Technology-based Intervention |
| VM | Video Modeling |
| VR | Virtual Reality |

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
