# Peer review of "Application of Augmented Reality Interventions for Children with Autism Spectrum Disorder (ASD): A Systematic Review"

_computers, doi:10.3390/computers12100215_

Round 1

Reviewer 1 Report

This paper's key finding is summarising and characterising different papers rather than reviewing the papers. A review should provide details of the key findings beyond the high-level outcomes of the papers. E.g. how the improvements in social-emotional reciprocity and emotional intelligence were observed? how the study's conclusion was drawn? The review needs to provide this contextual information for readers to form their opinions on whether this paper's observation is reasonable. 

Typos:

- Table 1 - PHR listed twice

- In many places, '-' was in the middle of the word (not on the line cut-off). E.g. in Table 3, "exclu-sively", "edi-tor". 

Author Response

Authors’ Response to Reviewers’ Comments

We would like to thank the reviewer for the careful review of our paper and useful comments and suggestions to improve the quality and presentation of the paper. We have thus accordingly modified the paper. The following is our response to the comments of the reviewer. We hope that the revision is satisfactory, and the paper would now be found suitable for publication.

Reviewer-1

This paper's key finding is summarizing and characterizing different papers rather than reviewing the papers. A review should provide details of the key findings beyond the high-level outcomes of the papers. E.g. how the improvements in social-emotional reciprocity and emotional intelligence were observed? How the study's conclusion was drawn? The review needs to provide this contextual information for readers to form their opinions on whether this paper's observation is reasonable.

Response: Thank you for your valuable inputs and insightful comments. We appreciate your thorough review of our paper. We would like to respectfully clarify that our paper extends beyond high-level outcomes in the discussion section. For instance, we have detailed key findings of AR interventions in a paragraph within the discussion section.

Please see pages 16-17 lines 267-285:

The majority of the included research suggests that children with ASD benefit from exposure to AR-based interventions. However, the majority of research utilized small sample sizes. Two studies [46,52] only comprised a single subject. Fourteen of the twenty-five studies had individuals under the age of ten. One study cited as [58] missed the number of participants. There was rarely any kind of comparison to either a group of healthy volunteers or patients who were receiving conventional therapies. If the control group does not exhibit the similar features or the same questionnaires are provided at distinct points in the intervention procedure, comparing the changes produced by an AR intervention could be challenging. Considering this, some of the findings from the research included in this study might be regarded as preliminary and having limited practical utility. Due to the limited sample size and lack of a control group, extending the results to the entire population impacted by this disorder is challenging. Another factor that makes it difficult to generalize the results is the sample's gender ratio. It is established that ASD affects boys more frequently than girls (the most recent research indicates a 4:1 ratio [2]). However, certain studies [46,52,60] are conducted exclusively with affected boys, which may limit the validity of the conclusions. Lastly, it should be mentioned that some research was conducted on children labeled with high-functioning autism or Asperger syndrome. Consequently, only results from this sub-sample should be evaluated, as they cannot be generalized to the remaining children with ASD.”

Another key finding centers on the exploration of AR's potential in enhancing social-emotional skills and daily living activities for children with ASD, providing adaptable learning environments.

Please see page 16 lines 258-266:  

“A few other studies [22,42] evaluated the feasibility of game content to assist autistic children with their social and emotional development.  Other studies have been based on areas related to training in activities of daily living skills, teaching language, education and improving attention to individuals with ASD.  Studies [41,48,51,55,62,63] demonstrate that using augmented reality can offer adjustable learning environments including videos, 3D images, and animation where people with special needs can engage in more fulfilling interaction activities.  Children and adolescents with ASD could learn basic human skills like tooth brushing and financial skills through AR intervention [44,47].”

However, in response to your feedback, we have made further revisions to our manuscript. The fourth column of Table 4 now contains information on how improvements in social-emotional reciprocity and emotional intelligence were observed in each study. It is to be noted that among the 25 selected papers, 18 of them reported notable enhancements in social-emotional reciprocity and emotional intelligence.

We believe these additions will enhance the context provided to readers, allowing them to better assess the observations. 

The following changes have been made in the revised manuscript in page 16, lines 249-261:

 “Social skills, the most obvious deficiency in children with ASD, were given the most attention in the examined AR study. Using a blend of real-world and virtual elements to replicate social environments allows for the training to occur in a safe, regulated, and customizable environment. For ASD treatment, the characteristics of this kind of intervention are particularly intriguing. In these articles, improvements in social-emotional reciprocity and emotional intelligence [28, 47, 50, 53, and 59] were observed significantly. It is important to highlight that among the 25 selected papers, 18 of them documented notable enhancements in social-emotional reciprocity and emotional intelligence. In Table 4, we have illustrated the study purpose of each study and its outcome. Additionally, these approaches assist in the development of important social abilities such as initiation of play, social response, and facial expression perception. The studies [26, 28] found that AR systems can positively improve understanding of social greeting behavior and learning appropriate responses to greetings.”

Typos:

- Table 1 - PHR listed twice

- In many places, '-' was in the middle of the word (not on the line cut-off). E.g. in Table 3, "exclu-sively", "edi-tor". 

Response: We have meticulously reviewed the paper and have removed all unnecessary hyphens and errors. Also, the manuscript is revisited to correct typos and grammar.

Reviewer 2 Report

The purpose of the presented article is to summarize the results of various studies in the field of using augmented reality methods for improving individuals' social skills of patients with autism spectrum disorder. The authors selected 25 studies, describing in the thesis the process of selecting individual articles. In principle, it is possible to agree with the selection of articles. As part of the inclusive criteria, the condition of the study was set in a scientific journal. For the purposes of the review article, studies presented at important conferences could also be included.

In the discussion section, the authors appropriately summarized the results of individual studies. They evaluated the individual studies rather critically in terms of their applicability in a broader context, while it is really difficult to extrapolate data from individual studies to the entire population of people with ASD, as these diagnoses require an individual approach to individuals.

The article appropriately shows what needs to be focused on in further studies carried out in the given area. For future comparison needs in the field of AR use, the authors could rather focus on comparing the AR technologies used, because some technologies within AR (even if it is questionable whether all technologies meet the requirements for AR) are incompatible with each other, especially from the point of view of user immersion.

The following errors need to be corrected in the text:

Line 54: add an explanation of the MAR abbreviation 

Line 57: ex -amining -> examining

Line 58: train -ing -> training

Table 1: definitions - last row change VVirtual -> Virtual

Line 99: fol -lows -> follows

Line 126: posi- tioning -> positioning

Line 136: add an explanation of the IDD abbreviation

Line 161: fol- lowing -> following

Figure 1: if possible, increase the font for better readability and adjust the arrows in the image

Table 3: edit the hyphens in the text

Refer to Figures and Tables in the text in the respective chapters

Line 220: add an explanation of the CASP abbreviation

Table 5 should be shorten just into studies with Assessment of Inclusion Criteria and Possible Covariates

Author Response

Authors’ Response to Reviewers’ Comments

We would like to thank the reviewer for the careful review of our paper and useful comments and suggestions to improve the quality and presentation of the paper. We have thus accordingly modified the paper. The following is our response to the comments of the reviewer. We hope that the revision is satisfactory, and the paper would now be found suitable for publication.

Reviewer-2

The purpose of the presented article is to summarize the results of various studies in the field of using augmented reality methods for improving individuals' social skills of patients with autism spectrum disorder. The authors selected 25 studies, describing in the thesis the process of selecting individual articles. In principle, it is possible to agree with the selection of articles. As part of the inclusive criteria, the condition of the study was set in a scientific journal. For the purposes of the review article, studies presented at important conferences could also be included.

Response: Thank you for your valuable feedback and suggestion regarding the inclusion of conference papers in our review article. We appreciate your thoughtful input.

In our current paper, we have chosen to concentrate exclusively on journal papers for several specific reasons. Firstly, we aimed to maintain a more focused scope in order to provide an in-depth analysis of the research published in peer-reviewed, indexed journal articles. Journal papers often undergo a more rigorous peer-review process compared to conference papers, which typically undergo a shorter review cycle. This approach allowed us to maintain a high level of methodological and scientific rigor in our review. Secondly, given the extensive body of literature in our research area, incorporating conference papers would necessitate a substantial increase in the number of studies to be reviewed.

In the discussion section, the authors appropriately summarized the results of individual studies. They evaluated the individual studies rather critically in terms of their applicability in a broader context, while it is really difficult to extrapolate data from individual studies to the entire population of people with ASD, as these diagnoses require an individual approach to individuals.

Response: As you correctly pointed out, the challenges associated with working with individuals with ASD stem from the unique and individualized nature of this condition. In response to your feedback, we have revised our discussion section to emphasize the need for an individualized approach when working with individuals with ASD. The following changes have been made in the revised manuscript in page 26, lines 361-368:

“While the review has summarized and evaluated various studies, it is crucial to acknowledge that the effectiveness of interventions can vary widely from person to person. This individual variation presents a significant challenge when attempting to extrapolate data from individual studies to the entire ASD population. Therefore, it is important to provide emphasize the need for personalized, client-centered approaches in the treatment and support of individuals with ASD. This approach may lead to a better understanding of ASD as a spectrum and promote interventions aimed at enhancing the well-being and development of individuals with ASD.”

The article appropriately shows what needs to be focused on in further studies carried out in the given area. For future comparison needs in the field of AR use, the authors could rather focus on comparing the AR technologies used, because some technologies within AR (even if it is questionable whether all technologies meet the requirements for AR) are incompatible with each other, especially from the point of view of user immersion.

Response: Thank you for highlighting the significance of our review in identifying key areas for further research in the field. We truly appreciate your feedback. 

As per your suggestion, we have incorporated a brief segment in discussion section that emphasizes the importance of comparing AR technologies. This addition aims to address the compatibility and immersion issues inherent in various AR technologies, providing valuable guidance for future research and practical applications.

The following changes have been made in the revised manuscript in page 26, lines 335-340:

“Based on the reviews of selected articles, it is evident that future research in the field of AR for individuals with ASD should pay closer attention to comparing the various AR technologies employed. Since AR encompasses a wide range of technologies, some of which may potentially have compatibility issues, particularly concerning user immersion. Therefore, it is advisable that the researchers should shed light on AR technologies on their relative effectiveness, usability, and potential limitations.” 

The following errors need to be corrected in the text:

Line 54: add an explanation of the MAR abbreviation 

Line 57: ex -amining -> examining

Line 58: train -ing -> training

Table 1: definitions - last row change VVirtual -> Virtual

Response: All abbreviations have been defined in Table 1, and specific explanations for Mobile Augmented Reality (MAR) can be found in line 55. The Intellectual Development Disorder (IDD) is elaborated in lines 135-136.

Line 99: fol -lows -> follows

Line 126: posi- tioning -> positioning

Line 136: add an explanation of the IDD abbreviation

Line 161: fol- lowing -> following

Response: The manuscript is revisited to correct typos and grammar.

Figure 1: if possible, increase the font for better readability and adjust the arrows in the image

Response: The font size is increased for better readability and the arrows are adjusted in the image.

Table 3: edit the hyphens in the text

Refer to Figures and Tables in the text in the respective chapters

Response: The tables provide in-depth comparisons among the papers, and your feedback on referring to relevant tables throughout other sections is highly appropriate. The following changes have been made in the revised manuscript in page 13 lines 215-222.

Most of the studies included in this analysis involved participants whose ages ranged from 6 to 20 years old.  Among the all studies, 10 studies had a mean age between 6 to 9 years, while seven studies had a mean age between 10 to 20 years only three studies had a mean age below 5 years, and two studies had a mean age above 20 years old. Table 5 represents the assessment in which the characteristics of the studies were performed. This focuses on the inclusion criteria and the possible covariates of the selected studies.  Furthermore, Table 8 cumulatively illustrates the key features of the included studies. It shows the software & hardware that are used also the assessment method followed by the authors.”

in page 14 lines 227-228, 

“The CASP checklist has 10 questions divided in three sections: Section A, B, and C. Table 6,7 illustrates the CASP.  The following questions of the three sections have assisted in assessing the quality of the papers

and in page 16 lines 248-258 to better facilitate the recommendations-

Social skills, the most obvious deficiency in children with ASD, were given the most attention in the examined AR study. Using a blend of real-world and virtual elements to replicate social environments allows for the training to occur in a safe, regulated, and customizable environment. For ASD treatment, the characteristics of this kind of intervention are particularly intriguing. In these articles, improvements in social-emotional reciprocity and emotional intelligence [28, 47, 50, 53, and 59] were observed significantly. In Table 4, we have illustrated the study purpose of each study and its outcome. Additionally, these approaches assist in the development of important social abilities such as initiation of play, social response, and facial expression perception. The studies [26, 28] found that AR systems can positively improve understanding of social greeting behavior and learning appropriate responses to greetings.”

Line 220: add an explanation of the CASP abbreviation

Table 5 should be shorten just into studies with Assessment of Inclusion Criteria and Possible Covariates

Response: The manuscript is revisited rigorously and made necessary corrections to the tables (Table-1, 3, 5, 6, 7). Additionally, we have removed 2 columns from Table 5. Furthermore, we have included an explanation of the CASP.

Reviewer 3 Report

I would like to congratulate the authors for a very good paper.

However there are some issues that I am asking authors to answer.

1. You have a lot of words that look like the following:

re-search, evalua-tion, fol-lows, etc...

Please correct all these mistakes since it's misleading the readers.

2. A few tables are not mentioned in the text. They are present in the paper but no reference is made to them. Please correct this issue and add a relevant text.

3. In Table 6 you have this table head: Did the review address a clearly focused question?

About which review are you talking about? Same question also for all other references to review.

Author Response

Authors’ Response to Reviewers’ Comments

We would like to thank the reviewer for the careful review of our paper and useful comments and suggestions to improve the quality and presentation of the paper. We have thus accordingly modified the paper. The following is our response to the comments of the reviewer. We hope that the revision is satisfactory, and the paper would now be found suitable for publication.

Reviewer-3

I would like to congratulate the authors for a very good paper. However, there are some issues that I am asking authors to answer.

Response: We sincerely appreciate your time and recognition of our work. Thank you for your thoughtful review. Detailed responses to each of your concerns have been prepared, and the necessary revisions have already been made to enhance the quality of our paper.

  1. You have a lot of words that look like the following:

re-search, evalua-tion, fol-lows, etc...

Please correct all these mistakes since it's misleading the readers.

Response: The reviewer is correct in pointing out that the presence of typos without proper line breaks can be misleading to readers. We have meticulously reviewed the paper and have removed all unnecessary hyphens and errors.

  1. A few tables are not mentioned in the text. They are present in the paper but no reference is made to them. Please correct this issue and add a relevant text.

Response: The tables provide in-depth comparisons among the papers, and your feedback on referring to relevant tables throughout other sections is highly appropriate. This issue has now been addressed in the revised manuscript and relevant text has been added to rectify it.

The following change has been made in the revised manuscript in page 08 lines 212-221 and in page 14 lines 225-229.

Most of the studies included in this analysis involved participants whose ages ranged from 6 to 20 years old. Among the all studies, 10 studies had a mean age between 6 to 9 years, while seven studies had a mean age between 10 to 20 years. Only three studies had a mean age below 5 years, and two studies had a mean age above 20 years old. Table 5 represents the assessment in which the characteristics of the studies were performed. This focuses on the inclusion criteria and the possible covariates of the selected studies.  Furthermore, Table 8 cumulatively illustrates the key features of the included studies. It shows the software & hardware that are used also the assessment method followed by the authors.”

“The CASP checklist has 10 questions divided in three sections: Section A, B, and C. Table 6, 7 illustrates the CASP.  The following questions of the three sections have assisted in assessing the quality of the papers

  1. In Table 6 you have this table head: Did the review address a clearly focused question?

About which review are you talking about? Same question also for all other references to review.

Response:

We appreciate your observation regarding the terminology used in Table 6. You are correct; it should not refer to "review" but instead to "paper". We apologize for any confusion, and this has now been corrected in the table to accurately reflect the content under consideration. Thank you for bringing this to our attention.

Reviewer 4 Report

The paper propose a complete systematic review about the use of AR for ASD.

The search  engine use are the most imporant ones and the metholodgy adopted to extrapolete important information is well described.

The filterting approch is well done and well discrebed. I do not have particular suggestions to improve the paper since it is already well done.

The paper can be accepted as it is.

Author Response

Authors’ Response to Reviewers’ Comments

We would like to thank the reviewer for the careful review of our paper and useful comments and suggestions to improve the quality and presentation of the paper. We have thus accordingly modified the paper. The following is our response to the comments of the reviewer. We hope that the revision is satisfactory, and the paper would now be found suitable for publication.

Reviewer-4

The paper propose a complete systematic review about the use of AR for ASD.

The search engine use are the most important ones and the methodology adopted to extrapolate important information is well described.

The filtering approach is well done and well described. I do not have particular suggestions to improve the paper since it is already well done.

The paper can be accepted as it is.

Response: We sincerely appreciate your positive feedback and the thorough evaluation of our paper. Your comments are valuable to us, and we are pleased to learn that you found the paper well done. Your acceptance of the paper as it is, is greatly appreciated.  Thank you for your time and consideration.

Round 2

Reviewer 1 Report

The revisions have addressed my comments very well. 

Author Response

Authors’ Response to Reviewers’ Comments

We would like to thank the reviewer again for the careful review of our paper and useful comments and suggestions to improve the quality and presentation of the paper. We have thus accordingly modified the paper. The following is our response to the comments of the reviewer. We hope that the revision is satisfactory, and the paper would now be found suitable for publication.

Reviewer-1

The revisions have addressed my comments very well. 

Response: We sincerely appreciate your positive feedback and the thorough evaluation of our paper. Your comment is valuable to us, and we are pleased to learn that you found the revisions well done. Thank you for your time and consideration.

Reviewer 3 Report

There are just two small issues to correct before fully accepting this paper.

1. The font in figures doesn't seem to be the correct one.

2. I would suggest numbering rows in all tables since otherwise future readers should start counting all rows which is confusing.

Author Response

Authors’ Response to Reviewers’ Comments

We would like to thank the reviewer again for the careful review of our paper and useful comments and suggestions to improve the quality and presentation of the paper. We have thus accordingly modified the paper. The following is our response to the comments of the reviewer. We hope that the revision is satisfactory, and the paper would now be found suitable for publication.

Reviewer-3

There are just two small issues to correct before fully accepting this paper.

  1. The font in figures doesn't seem to be the correct one.
  2. I would suggest numbering rows in all tables since otherwise future readers should start counting all rows which is confusing.

Response: We appreciate you taking the time to provide us with valuable feedback. This enabled us to improve this paper's professionalism and furnishing. The work we did in response to your review is explained below. Thank for your review we have made necessary changes in the figures and in the tables and provided row numbers to each one.

The manuscript is also revisited to correct typos and grammar.
